# Pre-Treatment of Starter Cultures with Mild Pulsed Electric Fields Influences the Characteristics of Set Yogurt

**DOI:** 10.3390/foods12030442

**Published:** 2023-01-17

**Authors:** Corinna Stühmeier-Niehe, Luca Lass, Miriam Brocksieper, Panagiotis Chanos, Christian Hertel

**Affiliations:** 1Department of Biotechnology, German Institute of Technologies (DIL), Professor-von Klitzing Str. 7, 49610 Quakenbrück, Germany; 2Elea Technology GmbH, Professor-von Klitzing Str. 9, 49610 Quakenbrück, Germany; 3Applied Life Sciences, Hochschule Emden-Leer, Constantiaplatz 4, 26723 Emden, Germany; 4Faculty of Agriculture, Rheinische Friedrich-Wilhelms-Universität, Regina-Pacis Weg 3, 53113 Bonn, Germany

**Keywords:** pulsed electric field, fermentation, oxidative stress, starter culture

## Abstract

The aim of this study was to investigate the effect of pulsed electric field (PEF) pre-treatment of a dairy starter culture of *Lactobacillus delbrueckii* subsp. *bulgaricus* LB186 and *Streptococcus thermophilus* ST504 on the fermentation and final product characteristics of set-style yogurt. The effects of PEF treatment parameters, voltage (4–20 kV), pulse number (20–80 pulses), frequency (1–21 Hz), and pulse (5–8 µs) width on pH development, cell counts, and proteolytic activity, as well as on texture and degree of syneresis in yogurt were investigated by use of a two-level full factorial design. Pulse frequency and pulse width had a significant effect on the yogurt stiffness (*p* < 0.05) and the interaction of voltage and frequency had a significant effect on both stiffness and proteolytic activity (*p* < 0.05). Further experiments confirmed that pre-treatment of the dairy culture with specific PEF parameters immediately before addition to milk could accelerate fermentation of, increase stiffness of, and reduce syneresis in the final yogurt. This effect of the PEF-pre-treated culture was partially retained even after flash-freezing and 14 days of storage of the culture at −20 °C. The effects were attributed to responses to oxidative stress induced by the PEF pre-treatment.

## 1. Introduction

Dairy products like yogurt are in high demand, due to their good nutritional values and sensory attributes [1]. Yogurt, as a fermented food, can vary in its organoleptic properties, which are important for product quality and consumer acceptance. These properties depend on different production factors, e.g., treatment of the milk or the performance of the bacteria used in starter cultures. For fermentation of set-style yogurt, *Lactobacillus delbrueckii* subsp. *bulgaricus* and *Streptococcus thermophilus* are used together in a collaborative relationship termed proto-cooperation [2]. Both microorganisms can grow individually in milk; however, in a mixture, *S. thermophilus* grows fast and produces several metabolites like pyruvic acid, formic acid, lactic acid, and CO_2_, which, among others, reduce the pH of the milk and boost the growth of *L. delbrueckii*. In turn, by producing peptides, free amino acids, and putrescine, *L. delbrueckii* stimulates the growth of *S. thermophilus*.

Pulsed electric fields (PEF) in the food industry can be described as the application of electric current, in the form of pulses, flowing through the product within a treatment chamber [3]. The technology originally found application in the non-thermal pasteurization of liquid foods [4], but gradually new applications evolved, like the extraction of intercellular components [5] and the potential fortification of low-fat foods with fat-soluble nutrients [6], based on the electroporation and disintegration effect of PEF. In the context of novel applications, mild PEF (≤1 kV/cm) application to microorganisms has been shown to cause changes in the metabolism and the metabolome. This has been demonstrated by, among other things, changes in performance and enzymatic activity [7,8,9,10], increases in acid tolerance [11], biosynthesis of exopolysaccharides [12,13], and acceleration of carbohydrate metabolism [14,15,16]. 

Although considerable data in the literature is available on the electrostimulation effect of PEF on microorganisms, the knowledge available for its applicability in industrial food production is scarce. This is especially true for, but not limited to, fermented dairy products. Reduction of the total fermentation time or changes in the rheological characteristics of the dairy products are of high interest for the industry, due to potential cost reductions associated with shortening of the fermentation process or with later reductions of losses in processing and/or transport of the product.

From the literature, it is difficult to establish a clear and unambiguous connection between mild PEF treatment of microorganisms and observed metabolic effects. For example, the effect of the application of mild PEF on the growth of *L. delbrueckii* subsp. *bulgaricus* has been found to be either beneficial or detrimental [11,17]. This difficulty probably has its grounds in the fact that the effect of application of PEF on microorganisms is subject to a multitude of processing, environmental, and microbiological factors which differ widely among experimental works. The range of PEF-process-related parameters, like field strength, treatment time, pulse width, frequency, and temperature differ widely among experimental works. Furthermore, PEF-equipment-related parameters, like continuous or batch treatment and chamber design, as well as microorganism-related parameters, like culture and treatment medium conditions, not only add to the degree of diversity of the experiments but also ought to be properly reported in order to facilitate comparison between experimental works [18].

Although there is evidence of the potentially beneficial effects of mild PEF on the performance of microorganisms, this influence has not been considered in the context of commercial applications. The selection of starter cultures for food fermentation has traditionally been based on the screening of a large number of strains for desirable traits, like the speed of acidification, the enzymatic activity, or the production of exopolysaccharides. The application of stress through the pre-treatment of starter cultures with PEF could provide an alternative method of producing starter cultures “tuned” to achieve a specific outcome which is desirable to the food industry. Furthermore, the relative ease at which a PEF unit might be integrated into an existing production line of fermented food for the pre-treatment of starter cultures could give the freedom to food manufacturers, especially small- to medium-sized ones, to alter the properties of their products or create new products in a simple and cost-effective way. 

The aim of this study was to evaluate the influence of mild PEF pre-treatment of a starter culture consisting of *L. delbrueckii* subsp. *bulgaricus* and *S. thermophilus* on their subsequent performance in yogurt fermentation as well as on some characteristics of the final product. Furthermore, the persistence of the altered performance of PEF-pre-treated culture after its prolonged frozen storage was investigated in the production of yogurt. 

## 2. Materials and Methods

### 2.1. Bacterial Strains and Culture Conditions

*Lactobacillus delbrueckii* subsp. *bulgaricus* LB186 and *Streptococcus thermophilus* ST504 were kindly provided by Sacco S.r.L., Cadorago, Italy. Stock suspensions were stored at −20 °C in mMRS broth [19] for *L. bulgaricus* and M17 broth (Thermo Fisher Diagnostics GmbH, Wesel, Germany) for *S. thermophilus,* supplemented with 50% glycerol. Overnight (16 h) cultures were prepared by inoculating 100 mL of mMRS broth or M17 broth in 250 mL flasks with *L. bulgaricus* LB186 or *S. thermophilus* ST504 and incubating at 37 °C or 42 °C respectively with agitation (50 rpm). Working cultures were prepared by inoculation of the respective media with overnight cultures to an OD_600_ of 0.4 and incubation for 8 h to obtain a final concentration of 3 × 10^8^ CFU/mL. Thereof, cell suspensions were prepared by centrifugation at 10,000× *g*, washing twice in sterile buffered peptone water (Carl Roth, Karlsruhe, Germany), and re-suspending in the same. The starter culture was created by mixing equal volumes of the OD-adjusted cell suspensions of each strain. 

### 2.2. Pulsed Electric Field (PEF) Treatment 

The PEF system PEFPilot^TM^ Dual (Elea Vertriebs- und Vermarktungs mbH, Quakenbrück, Germany) was used to treat the starter cultures in peptone water. The treatment chamber had a volume of approx. 70 mL, with 20 cm length, 5 cm width and 1 cm depth in the center (the bottoms of the 20 cm edges were rounded). The stainless-steel electrodes were situated at the sides of the 20 cm axis and were parallel to each other (Figure 1). 

The PEF system created monopolar pulses with rectangular decay. Four different parameters were varied, i.e., voltage (V), pulse number (n), frequency (Hz), and pulse width (µs), using a 2^4^ full factorial plan, supplemented with center points. The high- and low-level settings for each PEF parameter were coded as −1 and +1, while the center point settings were coded as 0 (Table 1). Voltage settings were set at either 4 kV (−1), or 20 kV (+1). Pulse number was set at 20 (−1) or 80 (+1) pulses. Frequency was set to 1 Hz (−1) or 21 Hz (+1), whereas pulse width was set to 5 µs (−1) or 8 µs (+1). Each experiment was performed in at least 3 independent replicates and experiments were completely randomized. The settings of the center point were voltage 12 kV, 50 pulses, frequency 11 Hz and pulse width 7 µs. The temperatures of the cell suspensions were measured before and after the treatment with PEF; however, the temperature rise was <1 °C in all cases. After treatment, the suspension was either used directly to inoculate milk, or centrifuged (1000× *g*, 10 min, 4 °C) and the tube containing the pellet was frozen by immersion in liquid nitrogen and stored at −20 °C until use.

The effects of the four parameters were evaluated on the acidification capacity, cell count, oxidation reduction potential, and proteolytic activity of the cultures, as well as the degree of syneresis and texture of the yogurt produced with them. The factorial regression analysis was done with Minitab^®^, v. 18.1.

### 2.3. Yogurt Fermentation

Milk was prepared using 9% (*w*/*v*) skim milk powder (SMP LH Basic skimmed milk powder, DMK GmbH, Zeven, Germany) and dH_2_O. The milk was preheated to 90 °C for 10 min and stored at 4 °C for 1 day until use. For each run, duplicate fermentations were performed, using 200 mL sterile urine cups with screw caps (O. Kohl Chemie-Pharma Laborbedarf, Ritterhude, Germany). A round opening was cut on the lid to let the pH electrode through and was fitted with gaskets to prevent the introduction of condensed water in the beakers. Each beaker, containing 147 mL of milk preheated to 42 °C, was inoculated with the PEF-treated or non-treated starter culture to reach a final cell count of 5 × 10^6^ CFU/mL. Incubation of the beakers was done in a water bath (LAUDA Eco RE 2025) at 42 °C for 8 h, and pH and temperature were monitored online using the iCinac new generation analyzer system (AMS Alliance, F-Frépillon, France). The positions of the beakers in the water bath were fully randomized on each experimental day in order to neutralize possible uneven heating effects. Timepoint t_0_ (t = 0 min) was defined as the timepoint at which the cell suspension was inoculated into the milk, followed by slow mixing for a further 2 min. For texture, proteolysis and syneresis measurements, additional beakers were prepared and incubated at 42 °C in a separate water bath in parallel.

### 2.4. Monitoring the pH Development of Yogurt

Each yogurt fermentation was done in duplicate containers. The pH values for each time point were averaged between the duplicates, and the pH change for each sample was calculated by subtracting the mean pH value of each point from the median t_0_ pH of all the samples of the day. The pH lag phase λ_pH_ and maximum pH change µ_max_ of each fermentation were determined by fitting the Gompertz or the Boltzman model (Equations (1) and (2)) [20,21] on the pH change data using the XLfit^®^ (IDBS) add-in for MS-Excel and by calculating the parameters A to D, which refer to the maximum pH value reached and the time in which the maximal pH is achieved (Equations (3)–(6)). The parameters A to D refer to the minimum value reached by the pH change (A), maximum value reached by the pH change (B), time value when slope value is max (C), and the slope-correlated value, which describes its behavior but without any physical meaning (D).

Gompertz model:(1)$$A\cdot e^{-e^{\left(B-\left(C\cdot x\right)\right)}}$$

Boltzmann model:(2)$$A+\frac{B-A}{1+e^{\frac{C-x}{D}}}$$

Equation for µ_max_ based on the Gompertz model:(3)$$\mu_{\max}=\frac{A\cdot C}{e}$$

Equation for λ based on the Gompertz model:(4)$$\lambda_{\mathrm{pH}}=\frac{\left(B-1\right)}{C}$$

Equation for µ_max_ based on the Boltzmann model.
(5)$$\mu_{\max}=\frac{B}{4\cdot D}$$

Equation for lambda based on the Boltzmann model.
(6)$$\lambda_{\mathrm{pH}}=C-\frac{B}{2\cdot \mu_{\max}}$$

### 2.5. Microbiological Analysis

Samples were taken at different time points from cell suspensions before and directly after PEF treatment as well as from milk after inoculation. mMRS (for *L. bulgaricus*) and M17 (for *S. thermophilus*) agar were used for determination of the viable cell counts. For the factorial regression experiments, the counts of non-injured cells were determined by plating samples on mMRS agar and/or M17 agar supplemented with 16% (*w*/*v*) and 7% (*w*/*v*) sodium chloride, respectively. These NaCl concentrations in the two agars were previously selected as the maximum non-inhibiting salt concentrations for non-PEF-treated cells (one way ANOVA, *p* < 0.05, data not shown). Plates were incubated at 37 °C (*L. bulgaricus*) and 42 °C (*S. thermophilus*) for 48 h.

### 2.6. Syneresis and Texture Analysis

Yogurt samples were stored at −20 °C in 15 mL Falcon tubes. After thawing for 3 h at room temperature, syneresis was evaluated by centrifugation at 500× *g* for 10 min at 4 °C and weighing of the separated whey. Results were expressed as the weight percentage of serum released by centrifugation [22,23].

For texture analysis, after the end of fermentation, the yogurt was kept in the beaker at 4 °C for a maximum of 14 days, before being measured with the TA.XT *plus* Texture Analyzer (Stable Micro Systems, Godalming, Surrey, UK), according to Amani et al. [22]. For each sample, the force of the penetration at a defined speed was measured in the 200 mL urine cups in which the yogurt had been prepared (diameter: 5.8 cm, height: 8 cm). A cylindrical probe (36 mm in diameter) with a flat base was pressed down on the surface of the yogurt gel at a speed of 0.5 mm/s, with 100× *g* force at a penetration depth of 10 mm. The stiffness of the yogurt was measured by the force at breaking of the yogurt gel and it was defined as the maximum force value (N) before the first sudden reduction in the curve measuring the resistance force versus the travel distance of the plunger.

### 2.7. Proteolytic Activity

The o-phthaldialdehyde (OPA) method was used to determine the concentration of primary amines in milk, as described by Spencer et al. [24] with modifications. Two and a half ml of yogurt was mixed with 1 mL of distilled water and transferred into test tubes containing 5 mL of 0.75 N trichloracetic acid (TCA) (Carl Roth, Karlsruhe, Germany), vortexed, and incubated for 10 min at room temperature. The solutions were filtered through Whatman^®^ qualitative paper, Grade 2 (Rundfilter, Cytiva, Marlborough, MA, USA), and analyzed spectrophotometrically at 340 nm. Non-inoculated, non-incubated milk was used as a blank. The release of α-amino acid in moles per liter was calculated from Equation (7), where ε is the molecular absorption coefficient (6000 M/cm), ΔA340 is the measured change of the absorption coefficient, and F the dilution factor corresponding to the assay procedure [24].

Equation for determining the concentration of free amino acids (M) based on the OPA method:(7)$$\alpha -\mathrm{amine}\;\mathrm{concentration}\;\left(M\right)=\varepsilon \cdot \Delta A340\cdot F$$

## 3. Results

### 3.1. Factorial Experiments

Yogurt fermentations were performed using differently PEF-treated starter cultures and their effects on the kinetics of fermentation and the rheological and fermentation characteristics of yogurt were monitored. The calculated means of the pH lag phase (λ_pH_) and maximum pH change (µ_max_) and the counts of surviving and non-injured starter cells, as well as the percentage of serum released, stiffness of the final yogurt, and concentration of primary amines are listed in Table 1 and Table 2.

Factorial regression analysis of the effect of PEF factors on the attributes of yogurt as well as the counts of the starter culture revealed that the type of PEF treatment had a significant effect on some of the attributes evaluated. The regression models fitted using the responses had a high goodness-of-fit in general, and the R^2^ adjusted to the number of factors was in general high (0.64–0.95). No apparent violations were observed with regards to the normality and homoscedasticity of the residuals, and the number of observations with large residuals in each model was kept under 8% of the observations. In any case, observations with high residuals did not alter the significance of the effects of factors.

None of the PEF factors investigated had a significant effect on λ_pH_, μ_max_, or the degree of syneresis in the final yogurt after 8 h of fermentation. The pulse frequency was the most influential (but marginally not significant) factor (*p* = 0.061), as an increase in the frequency tended to decrease λ_pH_. The interaction of voltage and pulse width was an influential factor for μ_max_, however only marginally significant (*p* = 0.067). At low pulse width, an increase in the voltage tended to increase the µ_max_, while the opposite was the case when the pulse width was high. The number of pulses applied was a marginally significant factor (*p* = 0.059) for the degree of syneresis of the yogurt at the end of 8 h, in that the application of a high number of pulses tended to reduce the % serum released from the yogurt after 8 h of fermentation.

On the contrary, treatment of the yogurt starter culture with PEF before milk fermentation had an obvious and significant effect on the stiffness of the yogurt produced. Factorial regression analysis showed that the interaction of voltage and frequency, the frequency itself, and the pulse width had a significant influence on the maximum force for initial penetration (stiffness) of the yogurt gel matrix (*p* < 0.05). Low pulse frequencies and longer pulses favored the production of hard yogurt. At the low frequency level, an increase in the voltage increased the stiffness of the yogurt. The cell counts of *L. delbrueckii*, but not those of *S. thermophilus,* at t = 0 min of the yogurt fermentation (after PEF treatment and inoculation in milk) were significantly affected by the field strength of the PEF treatment (*p* < 0.05). Interestingly, an increase in the field strength, within the range of values tested, led to an average increase in the *L. delbrueckii* counts of 0.2 log_10_ CFU/mL. The interaction of voltage and pulse duration had a significant effect on the counts of non-injured *S. thermophilus* cells. No influence of PEF on the cell count was observed for the non-injured cells.

Furthermore, increased proteolysis was measured after 3 h of yogurt fermentation when the PEF-pre-treated culture was used. The interaction of the voltage and frequency of the pulses had a significant effect on the proteolysis (*p* = 0.047). At a low frequency an increase in the voltage led to increased proteolysis, while at a high frequency an increase in the voltage had the opposite effect. However, none of the factors had a significant effect on the concentration of α-amines at 6 or 8 h of fermentation.

### 3.2. Effect of PEF Treatment on the Performance of the Starter Culture after a Freeze–Thaw Cycle

Four different PEF treatments were selected from the factorial experiments, which corresponded to those yielding starter cultures with which yogurt with the shortest lag phase (PEF 5), the lowest syneresis (PEF 6), one of the highest proteolytic activity (PEF 4) and one of the highest stiffness (PEF 2) was produced. Cells treated with these PEF conditions were either used to inoculate milk directly (PEFx-D) or following flash-freezing in liquid nitrogen and 2 weeks of storage at −20 °C (PEFx-FF). The yogurt fermentations produced with these cultures were compared with their respective controls; that is, non-PEF-treated cells (control) or cells non-PEF-treated but flash-frozen and stored at −20 °C for 2 weeks (Control-FF). The λ_pH_ and μ_max_ of these fermentations, as well as the development of *S. thermophilus* and *L. delbrueckii* subsp. *bulgaricus* counts in the fermentation, are displayed in Table 3. The syneresis, the stiffness, and the development of proteolytic activity in yogurts made with the different cultures are displayed in Table 4.

The *L. delbrueckii* subsp. *bulgaricus* and *S. thermophilus* culture treated with PEF 5 settings (PEF5-D) was confirmed to yield a fermentation with a shorter lag phase than that made with non-PEF-treated culture (control, paired *t*-test, *p* = 0.033) by an average of 22 min. Interestingly, the counts of *L. bulgaricus* in PEF 5-D were significantly lower in the milk at t = 0 of the yogurt fermentation than those in the control sample (paired *t*-test, *p* < 0.001) with a mean difference of 0.13 log_10_ CFU/mL between the two. Fermentations done with PEF 5-FF and PEF 2-FF cultures exhibited a longer λ_pH_ in comparison with Control-FF, although this difference was not significant (*p* > 0.05). In contrast, fermentations done with PEF 4-FF and PEF 6-FF cultures exhibited shorter λ_pH_, then Control-FF, although this was only significant for PEF 6-FF at the α = 0.1 level (paired *t*-test). 

The µ_max_ of fermentations done with PEF-treated cultures was on average higher than that of control fermentations, although paired *t*-tests revealed no significant differences (*p* > 0.05). Nevertheless, it should be noted that the pH change rate in yogurt fermentations produced with PEF 5-FF cultures was higher in all three replicates compared with Control-FF, and on average 0.287 pH units/ hour faster than the control (SD = 0.205). 

Interestingly, in the same fermentation, the counts of *S. thermophilus* right after inoculation of the milk (t = 0) with the starter culture appeared to be consistently higher than in the respective control (Control-FF), in all three replicates, by an average of 0.16 Log_10_ CFU/mL, and the difference was statistically significant (paired *t*-test, *p* = 0.008). In contrast, the counts of *L. bulgaricus* in PEF5-D samples at t = 0 appeared to be consistently and significantly lower than those of the control (mean difference: 0.13 Log_10_ CFU/mL, paired t-test, *p* = 0.001). Application of PEF treatment did not seem to affect the degree of cell loss of *S. thermophilus* due to flash-freezing, subsequent storage at −20 °C for 14 days, and thawing. However, PEF 5 pre-treatment on the starter culture increased the survival rate of *L. bulgaricus* through the freeze–thaw cycle by an average of 0.15 Log_10_ CFU/mL compared with Control-FF culture cells (paired *t*-test, *p* = 0.092).

Application of PEF pre-treatment to the starter culture resulted in a consistent decrease in the degree of syneresis in yogurts produced with these cultures compared with those made with non-PEF-treated starter cultures, regardless of the PEF treatment or of additional flash-freezing and storing of the cultures at −20 °C. This effect was not always significant; however, PEF 4-FF, PEF 5-FF, and PEF 6-FF yogurts exhibited significantly reduced syneresis (paired t-test, *p* = 0.023, *p* = 0.082 and *p* = 0.032 respectively) compared with Control-FF yogurts. Similarly, PEF 2-D yogurts exhibited significantly reduced syneresis compared with control yogurts (paired *t*-test, *p* = 0.023). This difference was not significant between PEF 2-FF and Control-FF yogurts (Figure 2).

In general, pre-treatment of the starter culture with PEF led to the production of stiffer yogurt, either when the cultures were used directly after PEF treatment or after intermediate flash-freezing and storage at −20 °C for 14 days. Significant increases in yogurt stiffness (paired *t*-test, *p* = 0.063) were achieved in PEF 5-D yogurts. On the contrary, a significant increase in yogurt stiffness (paired *t*-test, *p* = 0.041) was observed in PEF 2-FF yogurts but not in PEF2-D yogurts. However, both PEF 4-D and PEF 4-FF yogurts exhibited higher stiffness compared with control and Control-FF, respectively, irrespective of whether they were made with freshly PEF-treated cells or with PEF-treated and flash-frozen cultures (paired *t*-test, *p* = 0.016 and *p* = 0.05 respectively, Figure 3).

Proteolytic activity in the yogurt fermentation was in principle affected by the pre-treatment of the starter culture with PEF. However, this was significant only right at the beginning of the fermentation (t = 0) in most of the fermentations (Figure 4). In general, this effect was not carried over to PEFx-FF yogurt fermentations. Interestingly, though, in the case of PEF 2 pre-treatment of culture cells (Figure 4), the proteolytic activity in PEF2-FF yogurts was consistently higher compared with that in Control-FF yogurts, and at 6 h of fermentation (t = 6) this difference was significant (*p* = 0.027, Figure 4). 

## 4. Discussion

In this study, we initially applied a screening factorial design in order to detect the beneficial effects of mild PEF pre-treatment of a yogurt starter culture on its performance in the defined conditions of our experiments. By doing this, we showed that, within the value ranges of the parameters investigated, mild PEF pre-treatment of our starter culture can influence its performance as well as the characteristics of the yogurt. We confirmed that individual PEF parameters and/or their interactions have significant effects on these characteristics. For example, it could be postulated that pulse width and pulse frequency significantly affected the stiffness of the final yogurt (*p* < 0.05). Furthermore, this first screening phase allowed us to focus on specific sets of PEF treatment conditions and investigate their effects on yogurt in further experiments.

Pre-treatment of the starter culture with PEF under specific conditions (PEF5) led to a significant acceleration of acidification by reduction of the pH lag phase (λ_pH_) without a significant change in the maximum pH change (μ_max_). This finding agrees with our earlier results in the same type of yogurt [25], although the PEF treatment conditions applied to the yoghurt starter culture were different in that case. The acceleration of acidification may be the result of several mechanisms, possibly even interacting with each other. The slight change in the ratio of cocci to rods observed due to the PEF-induced decrease in the counts of *L. delbrueckii* subsp. *bulgaricus* may have influenced this, as has been shown before [26]. Additionally, non-lethal effects of electrical currents have been suggested to affect the growth and metabolism of microorganisms by nutrient uptake through reversible electroporation [27] and transient diffusion of ions and molecules [16,28] or by stimulation of metabolic cascades and pathways [29,30]. 

Indeed, discharge of high-voltage pulses in cell suspensions has been shown to cause formation of reactive oxygen species in the suspension as well as in the cytoplasm [31] and result in oxidative stress. In the mixed starter culture of *S*. *thermophilus* and *L. delbrueckii* subsp. *bulgaricus*, the former has been shown to have oxidative response mechanisms to alleviate the stress caused by reactive oxygen species (ROS) by deployment of enzymes like NADH oxidase (Nox) and NADH peroxidase, which reduce ROS to H_2_O_2_ and eventually to H_2_O [32,33]. Furthermore, oxidative stress due to increased dissolved oxygen (DO) concentration has been shown to lead to increased production of H_2_O_2_ by *L. delbrueckii* subsp. *bulgaricus* and eventual suspension of growth [34].

Interestingly, stimulation of Nox in *Streptococcus thermophilus* may accelerate the elimination of DO in a co-culture with *L. delbrueckii* subsp. *bulgaricus* and result in acceleration of fermentation manifested by an earlier pH decrease [33]. This may be explained by the fact that, at high oxygen concentration, the metabolism of *S. thermophilus* could shift from homolactic to mixed fermentation, with the production of end products like acetate, a-acetolate, acetoin, and diacetyl [32] effectively slowing down acidification. From the above, it is conceivable that, in our system, the application of PEF elicited an oxidative stress response on *S. thermophilus*, partly driven by the activation of Nox. When the culture was later introduced into the oxygenated milk medium, as in our case due to stirring, the oxidative stress response mechanism of *S. thermophilus* was presumably already activated, thus allowing the cells to cope more effectively with the presence of oxygen in the milk medium. This may have resulted in the shortening of the mixed fermentation phase due to faster elimination of DO in the medium and the return to a homolactic fermentation. In effect, this may have shortened the lag phase and accelerated the fermentation compared with the yogurt produced with non-PEF-pretreated cultures.

The increase in the stiffness of yogurts prepared with PEF-pre-treated starter culture, in conjunction with the concomitant reduction of syneresis, may indicate increased production of exopolysaccharides (EPS) [35]. Enhancement of EPS production by treatment of lactic acid bacteria with PEF has been shown before by Ohba et al. [12]; however, the underlying mechanism proposed, with concomitant reduction in the molecular weight of EPS compared with the control, is unlikely to contribute to the increase in the stiffness of yogurt [36]. Alternatively, Podolska et al. [36] observed an increase in the hydrophobicity of cells of *Lactobacillus plantarum* when treated with mild PEF (70 V/cm) for as long as 75 min. This increase in hydrophobicity was assumed to be caused by the production of EPS, as manifested by mucous exudate during PEF treatment, indicating a metabolic reaction of the cells to the treatment.

Indeed, the production of EPS by microorganisms has been known to occur as a response to different stress factors and is associated with oxidative stress resistance in LAB [37]. This is thought to be achieved, for example, by scavenging of ROS [38] or by extrusion of dissolved oxygen from the culture medium [39]. The mechanism by which the production of EPS in LAB is induced through the application of oxidative stress has not been fully understood yet; however, it is known that bacteria react to increased concentrations of cytoplasmic H_2_O_2_ by overproduction of EPS [40], and that the expression of oxidative stress response enzymes like NADH oxidase might be the key to this mechanism. In this context, the EPS production of *Lactobacillus casei* was found to be elevated dramatically by overexpression of a recombinant NADH oxidase [41]. This effect was coupled with a reduction in growth as well as a decrease in lactate production, which could be explained by the reduction in the NADH/NAD+ ratio leading to a decrease in the reduction rate of pyruvate to lactic acid. Taking this into account, it is likely that an overexpression of Nox in *S. thermophilus*, due to induction of oxidative stress through PEF treatment, may have caused a transient diversion of the metabolism to EPS production, while at the same time counteracting the induced oxidative stress by reducing intracellular and extracellular ROS. Whether such transient stimulation of EPS production alone would be sufficient to bring about the significant changes seen in the stiffness of yogurt and cheese made with PEF-pre-treated starter cultures is a matter for further research.

Pre-treatment of the dairy culture with PEF resulted in an immediate increase in the proteolytic activity at t_0_ (Table 4), measured as the concentration of primary amines in the milk medium. This increase was significant in most cases (*p* < 0.05 or *p* < 0.1, paired *t*-test). The development of the proteolytic activity after this point was very different among yogurt fermentations made with differently PEF-pre-treated starter cultures. Although *S. thermophilus* is generally poorly proteolytic, in our starter culture, both *Lactobacillus delbrueckii* subsp. *bulgaricus* LB186 and *Streptococcus thermophilus* ST504 exhibited weak proteolytic properties. An increase in the proteolytic activity of *L. acidophilus* and *L. delbrueckii* subsp. *bulgaricus* after mild PEF treatment has also been observed by Najim et al. [11]; however, this occurred gradually, reaching a maximum at 24 h of fermentation. In our work, given the short time between the PEF pre-treatment of the starter culture cells and freezing of the t_0_ sample of yogurt fermentation, it is questionable whether the observed immediate and significant increase in the concentration of primary amines could be a result of the activity of newly synthetized protease molecules as a response to PEF-induced stress.

An immediate increase in the activity of an enzyme due to PEF has been shown for pectinmethylesterase (PME) [42], possibly through activation of the enzyme by accumulation of cytoplasmic potassium on the cell membrane due to PEF-induced cell disruption. Indeed, application of PEF has also been shown to cause an accumulation of cations from the growth medium on the cell biomass [43]. Moreover, the activity of several peptidases and proteases in LAB have been shown to be dependent on different metal ions [44], including sodium and potassium, as shown before [45]. Therefore, the sudden increase in proteolytic activity in our work may have been the result of the accumulation of sodium and potassium ions from the buffered peptone water on the cells.

In addition, the application of PEF has been shown to have a direct effect on the activity of enzymes, which was dependent on the power of the PEF treatment as well as on the enzyme itself [46]. Immediate increases in the activity of various enzymes in the range of 5–20% were observed when the field strength was kept under a specific level, while over this level the opposite effect took place [46]. Interestingly, a recent study by Yun et al. [47] attributed the immediate increase in the activity of trypsin after application of PEF to alterations in the active center of the enzyme and to the increase in the affinity between enzyme and substrate. From our results, it is unclear whether any of the mechanisms described above may be contributing to this effect, and therefore further experiments may be needed for the elucidation of this mechanism.

In yogurt production, proteolysis, especially by *L. delbrueckii* subsp. *bulgaricus* in milk fermentation, increases the amount of free amino acids, which are essential for the growth of *S. thermophilus*. The proteolytic activity typically intensifies in the logarithmic growth phase and continues through storage at refrigeration temperatures. The production of amino acids also contributes to the taste of yogurt, either positively by acting as a precursor for the formation of volatiles or negatively by increasing bitterness through the accumulation of glutamic acid and proline [48]. Similar effects of excessive proteolysis have been observed in the taste of cheese as well [49].

In our work, although a high initial proteolytic activity was observed in yogurts produced with PEF-pre-treated cultures (PEF 2, PEF 5, PEF 6), no significant differences were observed towards the end of the fermentation, except for yogurts prepared with PEF-2-pre-treated cultures. The increased proteolytic activity at the beginning of fermentation did not seem to affect the growth of the starter culture, as indicated by the insignificant differences between the cell counts of PEF-pre-treated and control starter cultures at t = 3 h. However, the counts of PEF-2-pre-treated starter cultures at t = 3 h exhibited an increase of approximately 0.3 log_10_ CFU/g of *S. thermophilus* (Table 3), which might be attributed to the sustainably higher proteolytic activity of the PEF-treated culture on average compared with the control in the first 3 h of fermentation (Figure 4). Furthermore, it is doubtful whether the initially high proteolytic activity had any effects on the stiffness of yogurt, as no clear relations could be observed. Although some earlier research may suggest that EPS-producing starter cultures with high proteolytic activity could lead to softer yogurt [22], it seems that the proteolytic activity has not been shown to affect yogurt viscosity [50,51]. Moreover, the direct effect of proteolytic activity on the stiffness of the gel in set-style yogurts is largely unknown.

The ability of PEF-pre-treated cultures to bring changes to the fermentation and final product characteristics was, astonishingly, retained, to some extent, even after frozen storage of the pre-treated culture. To this extent, the use of a pre-treated culture, which had been stored frozen for 14 days, in yogurt fermentation retains its ability to increase the stiffness of yogurt and reduce the degree of syneresis compared with the same control culture stored frozen in the same way. In contrast, any effects on λ_pH_ and μ_max_ were abolished, while an increase in the proteolytic activity in PEFx-FF yogurts compared with Control-FF was observed mainly towards the end of the fermentation in some yogurts (Figure 4). 

The presumptive induction of oxidative stress on the starter culture cells through PEF did not seem to exert any cross-protective effect to the subsequent stress caused by freezing the cells, as derived from the cell counting (Table 3). Stress adaptation and cross-protection has been described in many prokaryotes, as well as specifically in LAB; however, adaptation to oxidative stress as a cross-protective measure to freezing has not been studied so far [52,53]. Furthermore, the fact that, in our work, PEF-pre-treated cells were immediately flash-frozen may have not allowed adequate time to completely develop the required cross-protective response.

The retention of the above-mentioned fermentation performance in the frozen cells may be of high importance to the starter culture industry. Alteration of the performance of starter cultures through their pre-treatment with PEF could open new possibilities to modify the performance of cultures as well as the properties of fermented milk products like yogurt, depending also on the genetically determined characteristics of the starter strains. PEF pre-treatment of the starter culture may thus be used to yield, for example, firmer yogurt, being more resistant to damage due to transportation. 

## 5. Conclusions

The application of PEF to yogurt starter culture as a pre-treatment to milk fermentation can alter its performance in milk fermentation and may be used as a means to modify the properties of the final product without changing the fermentation conditions or starter culture. The alterations brought about by the PEF pre-treatment of starter culture are dependent on the PEF process parameters and may be manifested as changes in the stiffness or the syneresis of yogurt, and may also have an effect on the sensory characteristics of the final product. Therefore, this technology can be of high importance to the dairy industry as a means to modify product characteristics, enhance the mechanical resilience of the products to transport, and reduce waste.

## 6. Patents

Part of the data has been used in a patent application. The patent application is currently under examination.

## Figures and Tables

**Figure 1 foods-12-00442-f001:**
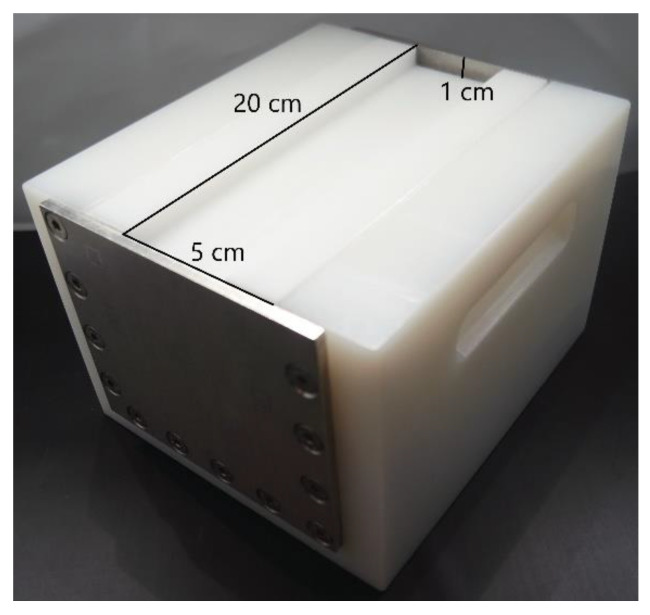
The PEF cell and the dimensions of the treatment chamber used.

**Figure 2 foods-12-00442-f002:**
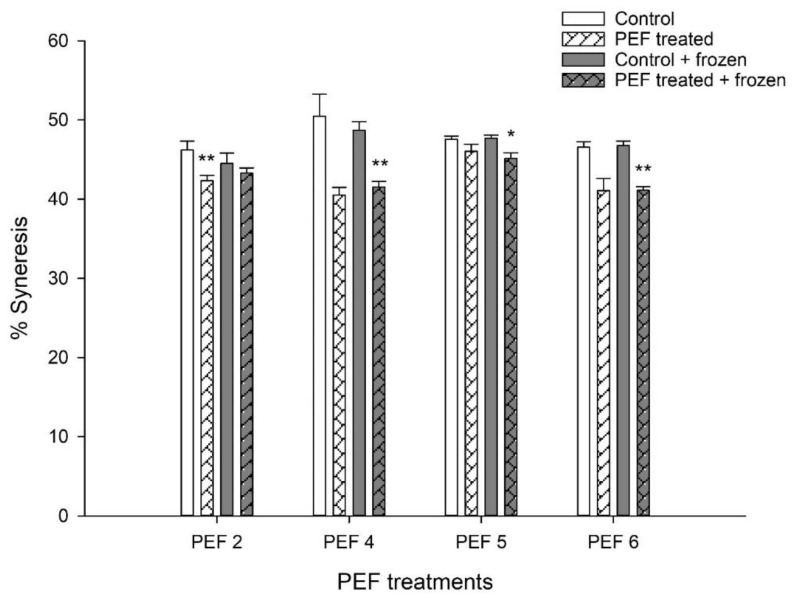
Effect of PEF pre-treatment of starter cultures on the syneresis of set yogurt. Yogurts were made with freshly PEF-pre-treated cultures (PEFx-D) or with cultures which were flash-frozen and stored at −20 °C for 14 days after PEF pre-treatment (PEFx-FF). PEF2, PEF4, PEF5 and PEF6 represent different PEF parameters. The bars indicate the mean values of three independent experiments ±1 standard deviation. * *p* < 0.1, ** *p* < 0.05 (paired *t*-test) compared with the respective controls (control vs. PEF-treated; control + frozen vs. PEF-treated + frozen).

**Figure 3 foods-12-00442-f003:**
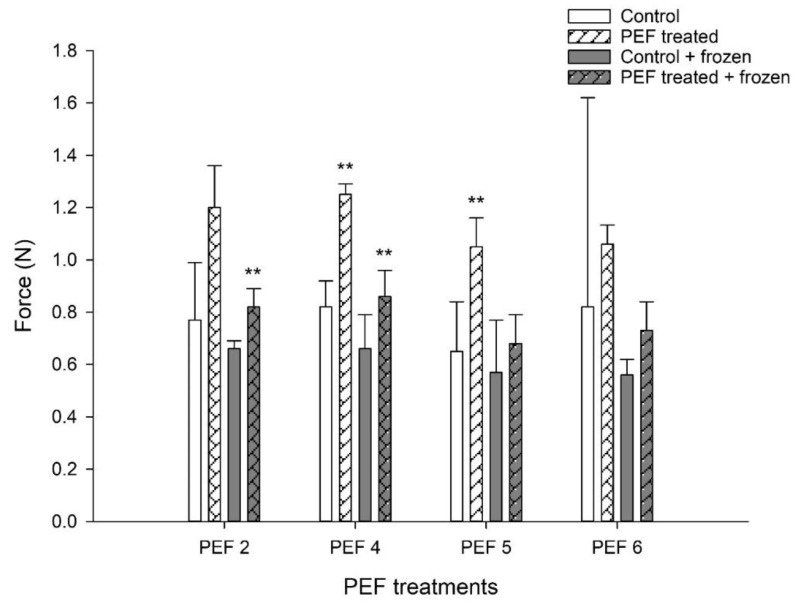
Effect of PEF pre-treatment of the starter culture on the stiffness of set yogurt. Force (N) is the maximum force applied on the surface of the yogurt until the first break. Yogurts were made with freshly PEF-pre-treated cultures (PEFx-D) or with cultures which were flash-frozen and stored at −20 °C for 14 days after PEF pre-treatment (PEFx-FF). PEF2, PEF4, PEF5 and PEF6 represent different PEF parameters. The bars indicate the mean values of three independent experiments ± 1 standard deviation. ** *p* < 0.05 (paired *t*-test) compared with the respective control (control vs. PEF-treated; control + frozen vs. PEF-treated + frozen).

**Figure 4 foods-12-00442-f004:**
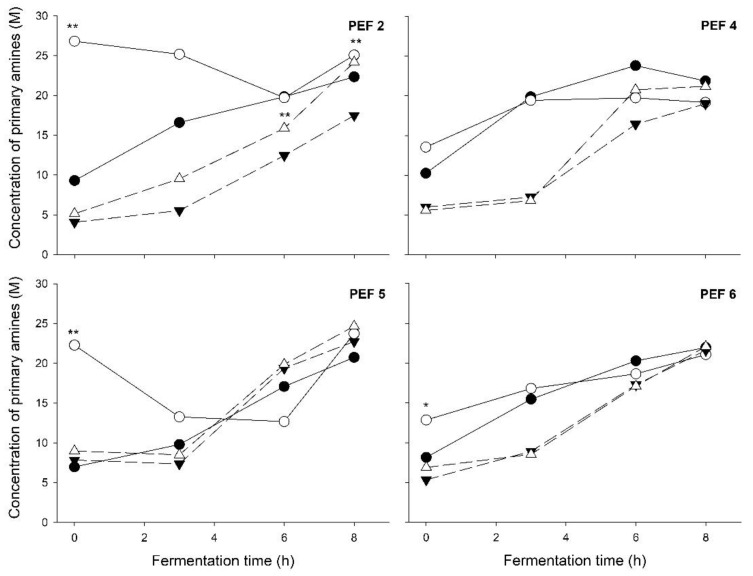
Proteolytic activity at different time points of yogurt fermentation made with differently PEF-treated starter cultures (PEF 2, PEF 4, PEF 5, PEF 6). ●: control; ○: PEF-treated; ▼: control + frozen; △: PEF-treated + frozen. Significant difference to the respective control (control vs. PEF-treated; control + frozen vs. PEF-treated + frozen; paired *t*-test), *: *p* < 0.1; **: *p* < 0.05. Error bars were removed for clarity.

**Table 1 foods-12-00442-t001:** Full factorial design with coded levels of PEF factors (voltage (V), pulse number, frequency (Hz), pulse width (µs)) for the yogurt fermentation experiments with PEF-treated starter cultures.

PEF Treatment	PEF Factors				Mean λ_pH_ ± SD (min)	Mean µ_max_ ± SD (pH/min)	Log10 Cell Counts (CFU/g)			
							*S. thermophilus* ± SD		*L. bulgaricus* ± SD	
	Voltage (V)	Pulse Number (n)	Frequency (Hz)	Pulse Width (µs)			SC	NI	SC	NI
1	−1	−1	−1	−1	78.66 ± 22.26	0.0128 ± 0.0008	8.20 ± 8.01	8.02 ± 7.76	8.22 ± 8.06	8.11 ± 7.93
2	−1	−1	−1	1	73.53 ± 13.47	0.0143 ± 0.0018	7.79 ± 7.68	7.68 ± 7.73	8.14 ± 7.97	8.12 ± 7.98
3	−1	−1	1	−1	64.36 ± 21.37	0.0144 ± 0.0017	8.13 ± 7.88	8.02 ± 7.78	8.19 ± 7.99	8.11 ± 8.00
4	−1	−1	1	1	65.68 ± 11.98	0.0152 ± 0.0004	8.26 ± 8.10	8.03 ± 7.73	8.25 ± 8.09	8.20 ± 8.10
5	−1	1	−1	−1	58.40 ± 8.31	0.0140 ± 0.0015	8.65 ± 8.60	8.11. ± 7.99	8.28 ± 8.02	8.10 ± 7.92
6	−1	1	−1	1	72.92 ± 9.10	0.0144 ± 0.0015	8.45 ± 8.36	7.96 ± 7.84	8.11 ± 8.05	8.08 ± 8.03
7	−1	1	1	−1	68.94 ± 12.28	0.0135 ± 0.0012	8.48 ± 8.50	7.83 ± 7.73	7.91 ± 7.84	7.94 ± 7.87
8	−1	1	1	1	69.43 ± 1.48	0.0139 ± 0.0009	8.48 ± 8.46	7.97 ± 7.86	8.71 ± 8.77	7.94 ± 7.63
9	1	−1	−1	−1	66.77 ± 11.96	0.0139 ± 0.0021	8.26 ± 8.08	8.03 ± 7.85	8.26 ± 8.10	8.21 ± 8.08
10	1	−1	−1	1	72.08 ± 6.83	0.0126 ± 0.0009	8.93 ± 9.00	7.99 ± 7.92	8.80 ± 8.85	8.10 ± 8.07
11	1	−1	1	−1	82.92 ± 17.89	0.0129 ± 0.0005	8.87 ± 8.92	7.96 ± 7.81	8.92 ± 9.02	7.98 ± 7.75
12	1	−1	1	1	71.60 ± 15.35	0.0132 ± 0.0022	8.21 ± 7.97	8.02 ± 7.76	8.23 ± 8.07	8.09 ± 7.96
13	1	1	−1	−1	68.29 ± 18.14	0.0158 ± 0.0027	8.93 ± 9.00	7.88 ± 7.78	8.15 ± 8.12	8.10 ± 8.07
14	1	1	−1	1	78.00 ± 9.18	0.0126 ± 0.0013	8.71 ± 8.74	8.05 ± 7.99	8.73 ± 8.76	8.07 ± 8.04
15	1	1	1	−1	70.66 ± 23.57	0.0135 ± 0.0017	8.45 ± 8.30	8.13 ± 8.03	8.21 ± 8.09	8.01 ± 7.85
16	1	1	1	1	65.61 ± 8.66	0.0127 ± 0.0008	8.57 ± 8.58	7.79 ± 7.46	7.96 ± 7.79	7.91 ± 7.83
Center Point	0	0	0	0	70.54 ± 14.29	0.0134 ± 0.0015	8.19 ± 8.16	7.90 ± 7.87	8.15 ± 8.11	8.27 ± 8.27

The table displays the results for mean pH lag phase (λ_pH_) and mean maximum pH change (µ_max_) *±* standard deviation as well as for log10 counts of surviving cells (SC) and log10 counts of non-injured cells (NI) of *L. delbrueckii* subsp. *bulgaricus* and *S. thermophilus* ± standard deviation (*n* = 3). For the center point *n =* 6. N/A: not applicable.

**Table 2 foods-12-00442-t002:** Mean values for syneresis (% weight), maximum force values (N), and α-amine concentrations (M) after 8 h of fermentation ± SD. (*n =* 3, *n =* 24 for center point).

PEF Treatment	Syneresis (% Serum Released ± SD)	Texture Analysis Force (N ± SD)	Proteolysis Concentration in Primary Amines (M ± SD)		
			t = 3 h	t = 6 h	t = 8 h
1	44.57 ± 7.54	0.85 ± 0.48	20.66 ± 18.14	19.18 ± 12.66	18.62 ± 11.68
2	41.35 ± 7.36	1.37 ± 0.25	13.64 ± 11.47	13.00 ± 9.41	15.02 ± 6.27
3	41.80 ± 11.87	0.81 ± 0.38	26.00 ± 19.95	20.60 ± 15.55	22.52 ± 7.05
4	42.32 ± 9.82	0.83 ± 0.45	28.30 ± 15.66	26.78 ± 16.13	29.32 ± 16.27
5	46.70 ± 18.30	1.48 ± 0.44	20.38 ± 9.59	23.70 ± 8.10	22.42 ± 16.36
6	38.93 ± 9.92	1.26 ± 0.06	13.60 ± 11.58	21.94 ± 7.15	14.62 ± 10.62
7	47.73 ± 16.55	1.02 ± 0.65	13.84 ± 13.86	11.16 ± 6.26	13.86 ± 9.41
8	43.58 ± 12.32	1.14 ± 0.06	39.86 ± 9.07	21.68 ± 14.99	26.42 ± 12.00
9	43.87 ± 6.63	1.21 ± 0.57	40.12 ± 12.37	29.44 ± 17.96	27.56 ± 13.19
10	39.39 ± 10.38	0.93 ± 0.18	32.06 ± 13.37	23.38 ± 12.99	45.26 ± 12.06
11	40.42 ± 15.86	1.50 ± 0.44	24.18 ± 20.51	22.08 ± 17.79	20.74 ± 15.43
12	49.40 ± 20.20	1.26 ± 0.36	23.92 ± 21.30	18.54 ± 13.83	20.82 ± 16.66
13	44.03 ± 9.11	1.05 ± 0.39	22.98 ± 17.23	20.86 ± 8.50	30.82 ± 11.67
14	46.55 ± 12.25	1.08 ± 0.30	30.56 ± 3.35	24.92 ± 10.62	36.32 ± 10.80
15	42.82 ± 11.68	1.12 ± 0.05	15.36 ± 16.80	22.04 ± 18.11	18.33 ± 16.23
16	36.78 ± 9.01	1.42 ± 0.47	21.88 ± 13.34	25.22 ± 17.61	26.32 ± 18.43
Center Point	43.67 ± 10.57	1.05 ± 0.39	23.26 ± 16.39	22.07 ± 14.43	25.61 ± 17.54

**Table 3 foods-12-00442-t003:** pH lag phase (λ_pH_), maximum pH change rate (μ_max_) and development of the cell counts of *S. thermophilus* and *L. delbrueckii* subsp. *bulgaricus* in yogurts made in with PEF-treated and non-PEF-treated cells (control), with (frozen) and without a subsequent freeze–thaw cycle (*n =* 3).

Starter Culture Treatment	Mean λ_pH_ ± SD (min)	Mean µ_max_ ± SD (pH/h)	Log10 CFU/g *S. thermophilus* ± SD				Log10 CFU/g *L. bulgaricus* ± SD			
			t = 0 h	t = 3 h	t = 6 h	t = 8 h	t = 0 h	t = 3 h	t = 6 h	t = 8 h
PEF 5										
control	94 ± 18.00	0.546 ± 0.025	5.98 ± 0.18	8.42 ± 0.17	8.59 ± 0.23	8.89 ± 0.16	5.91 ± 0.10	8.01 ± 0.29	8.49 ± 0.07	8.77 ± 0.19
PEF-treated	73 ± 11.00 **	0.597 ± 0.052	5.88 ± 0.12	8.38 ± 0.15	8.43 ± 0.08	8.86 ± 0.11	5.78 ± 0.09 ***	7.99 ± 0.37	8.36 ± 0.18	8.69 ± 0.24
control + frozen	188 ± 22.80	0.628 ± 0.140	5.04 ± 0.14	5.89 ± 0.36	8.28 ± 0.29	8.92 ± 0.06	5.02 ± 0.16	6.99 ± 0.26	8.45 ± 0.03	8.98 ± 0.21
PEF-treated + frozen	202 ± 30.60	0.915 ± 0.082	5.2 ± 0.17 ***	5.92 ± 0.39	8.32 ± 0.22	8.94 ± 0.14	5.03 ± 0.23	6.93 ± 0.31	8.43 ± 0.03	8.86 ± 0.03
PEF 2										
control	115 ± 23.90	0.536 ± 0.044	6.07 ± 0.12	7.72 ± 0.89	8.18 ± 0.11	8.83 ± 0.38	5.38 ± 0.75	8.23 ± 0.10	8.28 ± 0.14	8.52 ± 0.00
PEF-treated	109 ± 13.80	0.702 ± 0.101	5.88 ± 0.32	8.02 ± 0.10	8.19 ± 0.17	8.69 ± 0.41	5.33 ± 0.65	8.23 ± 0.06	8.15 ± 0.12	8.54 ± 0.44
control + frozen	185 (162–251) ^†^	0.703 ± 0.235	5.01 ± 0.06	6.51 ± 0.62	8.20 ± 0.38	8.60 ± 0.34	4.75 ± 0.26	7.08 ± 0.56	8.58 ± 0.09	9.08 ± 0.33
PEF-treated + frozen	196 (174–214) ^†^	0.718 ± 0.213	5.01 ± 0.06	6.79 ± 0.04	8.18 ± 0.36	8.74 ± 0.56	4.79 ± 0.22	7.20 ± 0.43	8.47 ± 0.35	9.10 ± 0.21
PEF 4										
control	95 ± 15.50	0.386 ± 0.031	6.06 ± 0.10	8.17 ± 0.33	8.35 ± 0.34	8.93 ± 0.16	6.09 ± 0.18	8.47 ± 0.32	8.73 ± 0.39	9.03 ± 0.13
PEF-treated	106 ± 6.70	0.447 ± 0.036	6.04 ± 0.10	8.18 ± 0.21	8.49 ± 0.22	8.91 ± 0.20	5.99 ± 0.06	8.47 ± 0.22	7.72 ± 1.56	8.81 ± 0.22
control + frozen	203 ± 6.90	0.601 ± 0.194	5.03 ± 0.11	6.29 ± 0.47	8.67 ± 0.32	9.06 ± 0.33	5.05 ± 0.14	6.87 ± 0.14	8.38 ± 0.57	9.06 ± 0.20
PEF-treated + frozen	182 ± 28.70	0.637 ± 0.142	4.93 ± 0.05	6.31 ± 0.52	8.54 ± 0.41	8.87 ± 0.18	5.01 ± 0.15	6.90 ± 0.19	8.06 ± 0.10	8.91 ± 0.08
PEF 6										
control	70 ± 9.97	0.544 ± 0.031	6.14 ± 0.16	7.99 ± 0.34	8.15 ± 0.23	8.83 ± 0.27	7.47 ± 1.27	8.27 ± 0.21	8.43 ± 0.26	9.01 ± 0.18
PEF-treated	69 ± 8.40	0.583 ± 0.055	6.10 ± 0.14	7.90 ± 0.29	8.12 ± 0.22	8.86 ± 0.31	6.05 ± 0.15	8.12 ± 0.13	8.47 ± 0.27	8.66 ± 0.42
control + frozen	146 ± 10.82	0.583 ± 0.139	5.12 ± 0.08	6.29 ± 0.18	8.17 ± 0.19	9.10 ± 0.18	5.13 ± 0.16	6.73 ± 0.24	8.46 ± 0.32	9.11 ± 0.21
PEF-treated + frozen	135 ± 6.38 *	0.614 ± 0.162	5.02 ± 0.07	6.27 ± 0.25	8.21 ± 0.10	8.96 ± 0.16	5.03 ± 0.12	6.76 ± 0.23	8.49 ± 0.33	9.13 ± 0.19

* Significant difference to control (*p* < 0.1); ** significant difference to control (*p* < 0.05); *** significant difference to control (*p* < 0.01). ^†^, Wilcoxon signed-rank test due to lack of normality.

**Table 4 foods-12-00442-t004:** Syneresis, yogurt stiffness, and development of proteolysis in yogurts made with-PEF treated and non-PEF-treated cells (control), with (frozen) and without a subsequent freeze–thaw cycle (*n =* 3).

Starter Culture Treatment	Syneresis (% Serum Released ± SD)	Texture Analysis Force (N ± SD)	Proteolysis Concentration in Primary Amines (M ± SD)			
			t = 0 h	t = 3 h	t = 6 h	t = 8 h
PEF 5						
control	47.57 ± 0.40	0.65 ± 0.19	6.98 ± 3.12	9.80 ± 7.61	17.08 ± 2.00	20.74 ± 3.70
PEF-treated	46.05 ± 0.91	1.05 ± 0.11 **	22.28 ± 5.35 **	13.26 ± 9.24	12.68 ± 8.60	23.74 ± 3.96
control + frozen	47.70 ± 0.41	0.57 ± 0.20	7.84 ± 5.16	7.36 ± 0.78	19.37 ± 2.19	22.74 ± 2.64
PEF-treated + frozen	45.15 ± 0.66 *	0.68 ± 0.11	8.98 ± 4.43	8.50 ± 0.81	19.86 ± 2.97	24.66 ± 3.19
PEF 2						
control	46.21 ± 1.13	0.77 ± 0.22	9.30 ± 4.58	16.60 ± 6.04	19.84 ± 9.74	22.34 ± 5.14
PEF-treated	42.35 ± 0.68 **	1.20 ± 0.16	26.82 ± 4.88 *	25.18 ± 12.61	19.72 ± 8.35	25.08 ± 5.87 **
control + frozen	44.52 ± 1.31	0.66 ± 0.03	4.08 ± 1.37	5.54 ± 2.71	12.46 ± 1.91	17.50 ± 5.23
PEF-treated + frozen	43.29 ± 0.71	0.82 ± 0.07 **	5.14 ± 1.08	9.54 ± 4.34	15.88 ± 0.92 **	24.18 ± 5.16
PEF 4						
control	50.48 ± 2.77	0.82 ± 0.10	10.26 ± 5.23	19.87 ± 11.13	23.79 ± 6.68	21.86 ± 5.24
PEF-treated	40.54 ± 0.95	1.25 ± 0.04 **	13.54 ± 3.33	19.42 ± 3.00	19.74 ± 0.57	19.16 ± 1.98
control + frozen	48.69 ± 1.09	0.66 ± 0.13	6.02 ± 0.52	7.26 ± 0.42	16.42 ± 2.34	19.00 ± 1.37
PEF-treated + frozen	41.55 ± 0.70 **	0.86 ± 0.10 **	5.60 ± 1.47	6.82 ± 0.71	20.76 ± 1.93	21.18 ± 3.58
PEF 6						
control	46.58 ± 0.70	0.82 ± 0.80 ^†^	8.14 ± 1.38	15.46 ± 3.17	20.28 ± 6.41 ^†^	21.94 ± 3.55
PEF-treated	41.10 ± 1.54	1.06 ± 0.07 ^†^	12.84 ± 2.59 *	16.8 ± 1.02	18.64 ± 2.64 ^†^	21.06 ± 4.67
control + frozen	46.79 ± 0.57	0.56 ± 0.06	5.32 ± 1.43	8.90 ± 1.18	17.26 ± 1.27	21.60 ± 1.75
PEF-treated + frozen	41.11 ± 0.47 **	0.73 ± 0.11	6.88 ± 1.04	8.52 ± 0.44	17.10 ± 1.48	22.08 ± 1.13

* Significant difference to control (*p* < 0.1); ** significant difference to control (*p* < 0.05). ^†^, Wilcoxon signed-rank test due to lack of normality.

## Data Availability

Data is contained within the article.

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
