# Peer review of "Pre-Treatment of Starter Cultures with Mild Pulsed Electric Fields Influences the Characteristics of Set Yogurt"

_foods, 2023, doi:10.3390/foods12030442_

Round 1
Reviewer 1 Report
Dear authors,
thank you for this relevant paper and idea. A lot of work has been done, and good luck with a patent application. Regarding manuscript:
-update reference (software issue)
2.2- Can you put a schematic setup of PEF device? Not sure how 70ml fits a 20 cm electrode gap? PEF parameters are not clear.
2.3. line118- preheated to 90 °C, for how long time?
Can you comment energy cost of this mild PEF treatment and all the mild benefits you have stated in this paper? How would this affect the final product price?
Reviewer 2 Report
I reviewed the manuscript entitled, Pre-treatment of starter cultures with mild pulsed electric fields influences the characteristics of set yogurt. The manuscript is well written and will contribute to the field. The structure of the manuscript is appropriate and sounds well. Based on these observations, in my opinion, the manuscript can be accepted for publication after addressing below points
Please remove the full stop from the title
What is set in title? I suggest to use yogurt
Abstract
Please provide little background on the study
Line 73: what is the mean of mild, what is the range?
Line 102: (Error! Reference source not found.)…. Please correct it
Please add ref for section 2.3 Yogurt fermentation
Equations: please write in an appropriate way. Use mathematical representation
Provide the detailed methodology for texture analysis
Line 208: Error! Reference source not found. and Error! Reference source not found.. ????
What are the mild conditions? Which should be mentioned in abstract
There is no standard for mild; authors should provide the conditions of PEF
Figure 1. there is no * in figure
In many places, I can see Error! Reference source not found. Authors should carefully revise the citations
Figure 3 quality should be improved
The discussion in some parts is very weak. Authors must consider discussing relevant latest literature.
References are not according to the journal format. Authors must perform extensive editing
